# Tracking of a Fixed-Shape Moving Object Based on the Gradient Descent Method

**DOI:** 10.3390/s22031098

**Published:** 2022-01-31

**Authors:** Haris Masood, Amad Zafar, Muhammad Umair Ali, Tehseen Hussain, Muhammad Attique Khan, Usman Tariq, Robertas Damaševičius

**Affiliations:** 1Electrical Engineering Department, Wah Engineering College, University of Wah, Wah Cantt 47040, Pakistan; haris.masood@wecuw.edu.pk (H.M.); tehseen.hussain@wecuw.edu.pk (T.H.); 2Department of Electrical Engineering, The Ibadat International University, Islamabad 54590, Pakistan; amad.zafar@iiui.edu.pk; 3Department of Unmanned Vehicle Engineering, Sejong University, Seoul 05006, Korea; umair@sejong.ac.kr; 4Department of Computer Science, HITEC University Taxila, Taxila 47040, Pakistan; attique.khan@hitecuni.edu.pk; 5College of Computer Engineering and Sciences, Prince Sattam Bin Abdulaziz University, Al-Kharj 11942, Saudi Arabia; u.tariq@psau.edu.sa; 6Department of Applied Informatics, Vytautas Magnus University, 44404 Kaunas, Lithuania

**Keywords:** object recognition, object tracking, gradient descent, particle filters

## Abstract

Tracking moving objects is one of the most promising yet the most challenging research areas pertaining to computer vision, pattern recognition and image processing. The challenges associated with object tracking range from problems pertaining to camera axis orientations to object occlusion. In addition, variations in remote scene environments add to the difficulties related to object tracking. All the mentioned challenges and problems pertaining to object tracking make the procedure computationally complex and time-consuming. In this paper, a stochastic gradient-based optimization technique has been used in conjunction with particle filters for object tracking. First, the object that needs to be tracked is detected using the Maximum Average Correlation Height (MACH) filter. The object of interest is detected based on the presence of a correlation peak and average similarity measure. The results of object detection are fed to the tracking routine. The gradient descent technique is employed for object tracking and is used to optimize the particle filters. The gradient descent technique allows particles to converge quickly, allowing less time for the object to be tracked. The results of the proposed algorithm are compared with similar state-of-the-art tracking algorithms on five datasets that include both artificial moving objects and humans to show that the gradient-based tracking algorithm provides better results, both in terms of accuracy and speed.

## 1. Introduction

Object recognition and tracking is still a major area of interest when it comes to digital image processing, pattern recognition, convolution neural networks and artificial intelligence [1]. The applications associated with object recognition range from surveillance [2], optical character recognition [3], human behavior detection [4], remote sensing [5], video activity localization [6], night-time vision [7] and biomedical image acquisition applications to deep learning techniques [8]. Although many applications have been developed thus far, the need for the optimization of the algorithms in terms of convergence and time minimization still persists. Object tracking particularly has many unique challenges associated with it, such as dealing with variations in scaling [9], occlusion [10], shift [11], camera axis orientations [12], etc.

Training tracking algorithms, such as approximate proximal gradient methods [13] and rapid gradient descent [14], is, in general, a very complex optimization issue because it involves a large number of secondary variables. Depending on the datasets and the problem at hand, the goal is to implement a tracking routine that provides faster results compared to its predecessors [15]. Besides the faster results, the algorithm should be accurate enough to concentrate on the object of interest only by minimizing the average tracking error. During the tracking routine, the tuning and assignment of weights are of utmost importance since they are the ones that mostly result in accurate prediction and estimation processes. For this purpose, a deep neural network technique known as gradient descent has been employed, which focuses on setting the weight of the parameters based on the lowest loss function.

Typically, object tracking is associated with several state-of-the-art techniques that are based on deep neural networks, artificial neural networks and convolution neural networks (CNNs) [16,17]. All of the aforementioned techniques have limitations ranging from poor interpretation and recognition of the object of interest to structural design issues. To solve these issues and to enhance the convergence of the algorithms, gradient descent training algorithms were proposed [18,19]. The gradient descent algorithms have the tendency to overcome most of the shortcomings of their predecessors by quickly converging into local minima but in an efficient manner [20].

One of the most famous types of gradient descent techniques is known as stochastic gradient descent (SGD). SGD is known to combine the benefits of basic gradient descent algorithms, i.e., the stochastic strategy and the backpropagation [21]. SGD can be used for image processing applications as it uses the backpropagation to converge quickly using local minima. This enables tracking algorithms such as particle filters to track the object in an efficient manner by quickly converging towards the object of interest [22,23,24,25].

In the previous decade, lots of visual tracking methods have been implemented, each having its share of pros and cons. The visual tracking methods can be categorized into two main classes, i.e., discriminative algorithms and generative algorithms. The generative methods are known to classify the object of interest by convolving it with a kernel first, followed by the tracking process that selects the most suitable candidate with an appearance model most suited to the chosen template. The most popular generative methods are particle-Kalman filters [26], Kalman filters [27], kernel-based object tracking [28], etc. On the contrary, the discriminative methods use binary classifiers in order to discriminate the object of interest from the background. The most popular discriminative methods employed thus far are ensemble tracking methods [29] and LDA and Bayes inference methods [30].

In [31], first, the dynamic behavior of the tracking model was assumed to be linear, which was used to model the motion of the objects using the parametric single acceleration method. The two sub-model states are estimated using an H filter. The estimates then act as input to the particle filters, resulting in the optimized state. The local estimates are mixed with the proposed interactive model for the calculation of the posterior location of the object of interest. Shi et al. [32] employed sparse representation for modeling of the target object. The target localization problem was assumed to be an L1 norm-related minimization problem and was resolved using convex optimization. The method was further improved [33] by proposing an lp regularization model. The lp regularization model was minimized using the accelerated proximal gradient approach, which ensured rapid convergence and less average tracking errors as compared to its predecessors [34]. The conditional random field (CRF) model has been used to combine multiple image texture, shape, context and location features for multiclass object recognition [35].

An eigenspace model for object tracking employs feature vectors linked with pixels in the target template, which are regarded as discrete observations of the target object [36]. To arrive at an eigenspace representation, the collection of observations is trained via non-linear subspace projection. A similarity function in the eigenspace representation is used to perform localization and segmentation. To optimize the similarity function regarding the transformation parameters, gradient descent and mean-shift approaches are used.

In this paper, an optimized algorithm for object recognition and tracking is proposed using SGD. The aim is also to develop a tracking routine that is able to track the object of interest under different scenarios. The proposed tracker will be tested on the target objects which undergo changes in appearance, scale and camera axis orientations. First, the object of interest is detected using the maximum average correlation height (MACH) filter. The object is detected using different parameters of the MACH filters. The detected object is then tracked in successive video frames using the SGD-based particle filters, which are enhanced forms of the conventional particle filter, providing better convergence than their predecessors. A comparison with similar state-of-the-art algorithms is performed to prove the effectiveness of the algorithm.

## 2. Proposed Methodology

The proposed methodology followed for the implementation of the algorithm is shown in Figure 1. The main algorithm can be split up into three parts, i.e., preprocessing, object recognition and object tracking. The trained images are kept in the library for object recognition purposes. Once the testing images are obtained, they are first fed to the preprocessing block to cater for any noise and/or any smoothing and sharpening abnormalities.

### 2.1. Preprocessing

Preprocessing is considered one of the most fundamental and very important steps in any image processing application. The step ensures that all the images possess similar dimensions and properties before the actual algorithm is applied to them. For preprocessing of the images, the difference of Gaussian (DoG) has been applied before the actual preprocessing of the images. The DoG filter not only reduces the noise by applying the Gaussian motion blur but also enhances the edges, which can be considered a major advantage when it comes to an image processing application [37]. The DoG simply is a two-step process. First, the DoG performs edge detection by applying the Gaussian motion blur. The motion blurring allows the removal of any unwanted noisy components by applying the smoothing process. The algorithm then applies another motion blur using a sharper theta. As the name implies, the final image is made by replacing each image pixel with the difference of blurred images as shown in Figure 2.

The DoG is a bandpass filter that is used to compute wavelets that are symmetric in nature. By changing the standard deviation in the equation of DoG, the bandpass frequency can be altered. The value of the bandpass frequency must be selected so that it can provide the best tradeoff between intra-class distortion and inter-class discrimination. Combining DoG with the MACH filter results in much sharper correlation peaks because of the built-in tendency of DoG to detect edges. The DoG filter enhances the edges by approximating the Mexican hat wavelet. The DoG is actually a variance between two scaled Gaussian functions $g_{i}\left(x,y\right),$ where i=1,2,… [38]:(1)$$g_{1}\left(x,y\right)\;=\frac{1}{2\pi \delta_{1}{}^{2}}\cdot \exp(-\frac{x^{2}+y^{2}}{2\cdot \pi \cdot \delta_{1}{}^{2}})$$

Similarly, for i=2,
(2)$$g_{2}\left(x,y\right)\;=\frac{1}{2\pi \delta_{2}{}^{2}}\cdot \exp(-\frac{x^{2}+y^{2}}{2\cdot \pi \cdot \delta_{2}{}^{2}})$$

The DoG filter is applied by taking the difference between Equations (1) and (2).
(3)$$g\left(x,y\right)\;=g_{1}\left(x,y\right)-g_{2}\left(x,y\right)$$

By combining Equations (1)–(3), we obtain
(4)$$g\left(x,y\right)\;=\frac{1}{2\pi \delta_{1}{}^{2}}\cdot \exp(-\frac{x^{2}+y^{2}}{2\cdot \pi \cdot \delta_{1}{}^{2}})-\frac{1}{2\pi \delta_{2}{}^{2}}\cdot \exp(-\frac{x^{2}+y^{2}}{2\cdot \pi \cdot \delta_{2}{}^{2}})$$

The rule of thumb for a successful DoG application is to choose the bandpass of the Mexican wavelet as the ratio of $\delta_{1}$ and $\delta_{2}$. Experimental results have shown that DoG yields closest approximation if the ratio of $\delta_{1}$ and $\delta_{2}$ is 1.6.

### 2.2. Object Recognition

Object recognition has been performed using the MACH filter, which is a class of correlation filters. The identification of an object is a straightforward approach, but identifying an object in the case of change in camera axis orientations or in the presence of noise is a challenging task. The correlation filters are chosen because of their ability to detect and identify the object of interest under challenging circumstances. Another advantage of correlation filters is they are computationally less expensive as compared to their counterparts. The MACH filter uses a set of training images for the computation of correlation peaks. The MACH distinguishes the training images into true and false classes, where the true class of training images depicts the set of images that the filter retains, while the false class represents the discarded set of images.

The MACH filters are known for their ability to suppress noise as well. The noise suppression ability comes with their ability to maximize average similarity measure (ASM) and reduce distortion. The MACH filter is based on enhancing the four main parameters, i.e., average correlation energy (ACE), average similarity measure (ASM), output noise variance (ONV) and average correlation energy (ACE). However, the most important aspect is the ASM, which is directly associated with the correlation peak. As mentioned above, the energy equation associated with MACH is based on four parameters.
(5)$$E\left(f\right)\;=\alpha \left(ONV\right)\;+\;\beta \left(ACE\right)\;+\;\gamma \left(ASM\right)\;-\;\delta \left(ACH\right)$$

For the MACH implementation, Equation (5) needs to be minimized. The “T” sign in superscript represents the transpose of the function
(6)$$\alpha f^{T}Cf+\beta f^{T}D_{x}f+\gamma f^{T}S_{x}f-\delta f^{T}m_{x}$$

As stated earlier, MACH is known to recognize the object of interest by minimizing the ACE and ASM while maximizing the average correlation height. The minimization of ASM is achieved using Equation (7). “*” represents the complex conjugate of the function.
(7)$$ASM=h^{+}\left[\frac{1}{N}\overset{N}{\underset{i=1}{\sum }}{\left(X_{i}-\bar{X}\right)}^{*}\left(X_{i}-\bar{X}\right)\right]h=h^{+}S_{x}h$$
where the similarity of the training images is represented by the matrix S. Similarly, the minimization of ACE is achieved using Equation (8)
(8)$$ACE=h^{+}\left(\frac{1}{N}\overset{N}{\underset{i=1}{\sum }}X_{i}X_{i}^{*}\right)h=h^{+}D_{x}h$$
(9)$${\left|\bar{u}\right|}^{2}={\left|h^{+}\bar{x}\right|}^{2}=h^{+}{\bar{\mathrm{xx}}}^{+}h$$

The average correlation intensity, which is depicted by ${\left|\bar{u}\right|}^{2}$, is maximized by the filter *h*. The same *h* is also used for the minimization of ASM and ACE.

While minimizing Equation (5), Equation (10) can easily be extracted after calculating the ASM and ACE.
(10)$$f=\frac{m_{x}^{*}}{\alpha C+\beta D_{x}+\gamma S_{x}}$$

The three parameters mentioned in the denominator of Equation (10) are considered crucial in the performance of MACH. The optimal value selection of these parameters is called optimal tradeoff (OT), which was initially proposed by Bone et al. [39]. The OT values of the parameters are based on two parameters, i.e., peak-to-correlation energy (PCE) and correlation output peak intensity (COPI) [40,41]. The PCE and COPI are calculated using Equations (11) and (12).
(11)$$COPI=\max\{|C(x,y){|}^{2}\}$$
(12)$$PCE=\frac{COPI-\bar{{\left|C(x,y)\right|}^{2}}}{{\left\{\sum \frac{{\left[{\left|C(x,y)\right|}^{2}-\bar{{\left|C(x,y)\right|}^{2}}\right]}^{2}}{N_{x}N_{y}-1}\right\}}^{1/2}}$$
where $\bar{{\left|C(x,y)\right|}^{2}}=\sum {\left|C(x,y)\right|}^{2}/N_{x}N_{y}$ is the average correlation output intensity.

Bone et al. [39] suggested that the values of 0.01, 0.1 and 0.3, respectively, for *α*, *β* and *γ* for the COPI cost function may be considered optimal.

Figure 3 shows a Blurred Vehicle along with the MACH filter results. It is pertinent to note here that even though the vehicle is undergoing severe blurriness, MACH is still able to detect the object of interest using the PCE and COPI indexes. The PCE and COPI values achieved for the object mentioned in Figure 3 are 2.3047 × 10^−5^ and 32.1376.

As mentioned earlier, MACH has the ability to detect the object of interest even if it undergoes changes in scaling, shifting or camera axis orientations. Figure 4 shows a vehicle traveling at night, and it has been scaled out by a factor of 3. MACH can still extract the correlation peak with a PCE value of 29.21.

Figure 5 shows a dog in a running mode along with its MACH results. The dog is the object of interest, which needs to be detected using MACH. The dog is currently encountering in-plane rotation and is not showing its original physical attributes. Even in such a tricky case like this, the MACH can detect the object of interest using a PCE value of 66.01.

Figure 6 shows a partially occluded case of a vehicle traveling on a road along with its MACH results. Besides the natural attributes of MACH providing invariance against shift, scales and camera axis orientations, it also possesses the ability to detect an object even it is partially occluded. Figure 6 shows that even the vehicle is partially occluded, MACH is still able to detect the object using PCE and COPI values of 68.5701 and 0.0014, respectively. The object of interest in this case was 30% occluded, and MACH still detects it successfully.

Once the object of interest is detected using MACH, a bounding box is used to encapsulate the object of interest, as mentioned in Figure 7. The coordinates of the bounding box are then fed to the gradient descent-based object tracking routine, which will update the bounding box with respect to the apparent motion of the object.

### 2.3. Object Tracking

The goal of object tracking is to track the recognized object of interest in successive frames. The main object tracking algorithm employs the gradient descent-based particle filters. The gradient descent technique allows the particle filters to track the object of interest in less time as compared to the conventional particle filters.

#### Gradient Descent-Based Particle Filtering

Gradient descent technique is one of the most famous methods used for optimizing algorithms. They have the natural tendency to converge when used together with other deep learning-based algorithms. In this paper, the gradient descent technique is used in conjunction with particle filters in order to improve the efficiency of the particle filters. The gradient descent technique optimizes the algorithms by working on their loss functions. The loss function can be described as the apparent difference between the function output and the samples. For this purpose, a hypothesis function can be assigned to a linear regression mathematical model, such as $h_{\theta }\left(x\right)=\theta_{0}+\theta_{1}x$, where $\theta_{0}\;\mathrm{and}\;\theta_{1}$ defines the equation factors. The samples can be defined as vector $\left(x_{i},y_{i}\right)=\left(1,2,\ldots ,n\right),$ such that every input $x_{i}$ corresponds to an output $y_{i}$. The loss function generally can be defined using Equation (13).
(13)$$J\left(\theta_{0},\theta_{1}\right)=\overset{m}{\underset{i=1}{\sum }}{\left(h_{\theta }\left(x_{i}\right)-y_{i}\right)}^{2}$$

The optimization of the loss function and the mathematical model is the main goal of the gradient descent technique. This means that the gradient descent technique can be used to amend or modify the mathematical model of a particle filter by reducing the eventual loss function.

In the case of particle filters, the hypothesis function can be very complex. Therefore, there is a need to define a more complicated hypothesis function. Adding more factors to the hypothesis function makes it more complex such that $h_{\theta }\left(x_{1},x_{2},\ldots x_{n}\right)=\theta_{1}+\theta_{1}x_{1}+\cdots +\theta_{n}x_{n}$. The loss function defined in Equation (13) can be made more complex using Equation (14).
(14)$$J\left(\theta_{0},\theta_{1},II,\theta_{n}\right)=\frac{1}{2m}\overset{m}{\underset{i=0}{\sum }}{\left(h_{\theta }\left(x_{0},I\ldots ,x_{n}\right)-y_{i}\right)}^{2}$$

To minimize the cumulative loss function based on the ensemble of particle filters and the gradient descent technique, partial derivatives can be applied. Gradients are generically applied for measuring the trend of the aim function. Therefore, it is possible to associate the $\theta_{i}$ result with the loss function. Equation (15) shows how a loss function can be represented using a gradient.
(15)$$\frac{\partial }{\partial \theta_{i}}J\left(\theta_{0},\theta_{1},\ldots ,\theta_{n}\right)$$

The precision of the function, i.e., the difference between the samples and the mathematical model, is represented by ϵ, which is known as the terminal function. The estimation process of particle filters will stop when the estimated difference becomes less or equal to ϵ. The particle filters are based on the prediction, estimation and upgrade regimes. The percentage of gradient used to upgrade the state of particle filters is determined by the controlling parameter α, which is the step size. The amended update expression for particle filters becomes:(16)$$\theta_{iold}-\alpha \frac{\partial }{\partial \theta_{i}}J\left(\theta_{1},\ldots ,\theta_{n}\right)$$

Since the function pertaining to the particle filters is convex in nature, it can be optimized using gradient descent. Another important aspect is the selection of the step size. Selecting a smaller step size may result in identifying the most optimal solution, but the downside is a very slow convergence speed. On the contrary, a larger step size may result in an elevated speed but does not ensure an optimal solution.

Considering the pros and cons of both approaches, in this paper, the gradient descent technique is used for updating the probabilities and assigned weights to the particles using the steps mentioned in Algorithm 1. The conventional particle filtering process involves the states $p\left(X_{n}|Y_{1:n}\right)$. We employ particle filters in combination with the gradient descent technique because the particle filters can work on both the linear and nonlinear systems, while gradient descent allows the particles of the particle filters to converge quickly. The quick convergence allows a much quicker tracking process as compared to the conventional particle filtering routines. Gradient descents’ ensemble with the particle filters starts by assigning a weight to the samples.

**Algorithm 1:** Gradient Descent Algorithm
Initializing the aim function parameters: $\theta_{0},\;\theta_{1},\;\ldots \theta_{n}$;Initialize step size α and terminal distance of recursion parameter *ϵ*;Gradient calculation of the function using the loss functions’ partial derivative using Equation (15);Apply the gradient descent algorithm on particle filters;If the gradient descents, i.e., $J(\theta_{0},\;\theta_{1},\;\ldots \theta_{n})\leq \epsilon$, stop the converging process;Otherwise, continue the process;Employ the gradient descent process by multiplying the step size α with the gradient;Renew all the values of *θ* using Equation (17).


Using the steps (5) and (6) of the gradient descent algorithm, new weights are assigned to the particles using Equation (17).
(17)$$W_{n}^{i}=W_{n-1}^{i}\frac{p\left(Y_{n}|X_{n}^{i}\right)p(X_{n}^{i}|X_{n-1}^{i})}{\Pi \left(X_{n}|X_{n-1}^{i},Y_{n}\right)},\overset{N}{\underset{i=1}{\sum }}W_{n}^{i}=1$$

Equation (17) shows the working particle filters after the implementation of gradient descent. The particle filters are based on two steps, i.e., prediction and then update. Equation (17) basically merges the two steps. The part of the equation involving the $p\left(Y_{n}|X_{n}^{i}\right)$ is used for the implementation of the probabilistic model. Once the probabilities are calculated, the weights of particle filters are set according to the expression $\left\{X_{n-1}^{i},W_{n-1}^{i}\right\}$. However, in contrast to the conventional particle filters concept, in this paper, the gradient descent technique is used to update the weights according to Algorithm 1.

## 3. Results and Discussion

Five different data sets have been used for testing the results of the gradient-based tracking routine. Since the gradient descent technique is known as the optimization algorithm, the results of the proposed algorithm will be compared with other state-of-the-art algorithms.

### 3.1. Data Sets

The first employed data set is the Blur Car, which shows a white vehicle moving on a road [42]. The data set is challenging in the sense that it possess challenges related to camera axis orientations as well as blurriness of the object of interest. Figure 8 shows the results of applying the gradient descent-based tracking routine. The tracker successfully tracks the vehicle in successive frames.

The second data set is Running Dog, which shows a dog running on a floor [43]. The data set is challenging in the sense that the object is changing shape. In addition, the data set also shows blurriness of the object of interest. Figure 9 shows the results of applying the gradient descent-based tracking routine. The tracker successfully tracks the vehicle in successive frames.

The third data set is Vehicle at Night, which shows a vehicle travelling at night [44]. The data set is challenging in the sense that the object is traveling at night under poor illumination conditions. In addition, the data set also shows blurriness of the object of interest. Figure 10 shows the results of applying the gradient descent-based tracking routine. The tracker successfully tracks the vehicle in successive frames.

The fourth data set is a grayscale vehicle traveling on the road [43]. The data set is challenging in the sense that the object becomes occluded under a bridge. In addition, the data set also shows blurriness of the object of interest. Figure 11 shows the results of applying the gradient descent-based tracking routine. The tracker successfully tracks the vehicle in successive frames.

The fifth data set utilized is called “Singer” [13], and it contains aphotographs of a singer performing in a concert. The dataset is deemed significant since the pictures are continually zoomed in and out, posing a challenge to the object tracking system, as illustrated in Figure 12.

### 3.2. Discussion and Comparison

To prove the efficiency of the algorithm with other state-of-the-art similar algorithms, the gradient-based tracking technique has been compared with four other recently proposed techniques. The algorithm has been compared with a target tracking algorithm based on Convolution Neural Network (TTACNN) [44], ADT: object tracking algorithm. Based on adaptive detection [45], vehicle tracking algorithm combining detector and tracker (VTACDT) [46], multi-object tracking for urban and multilane traffic (MTUMT) [47], adaptive weighted strategy and occlusion detection mechanism (AWSODM) [48] and approximate proximal gradient-based correlation filter (APGCF) [13]. Table 1 shows the execution times of these algorithms, while Table 2 represents the average tracking errors.

The comparison between different state-of-the-art recent algorithms was performed in terms of execution time (in seconds), average tracking errors, precision, mean average precision (MAP) and the precision recall for a minimum of 300 frames of five different image processing datasets. The results depicted in Table 1 clearly show that the proposed algorithm performs better in terms of execution time compared to its counterparts. The gradient descent approach manages the convergence of the particles in a fast time compared to the other tracking algorithms, thus providing a better execution time. In Table 2, the proposed algorithm encompasses much fewer average tracking errors compared to its counterparts. The average tracking error is measured by measuring the average deviation of the bounding box from its mean position. Table 3 shows the comparison of algorithms based on precision. Table 4 shows the comparison of the algorithms based on MAP, while Table 5 shows the comparison of the algorithms based on precision recall. All three metrices are precision-based and are measured using the position of the bounding box over the object of interest. Precision is the ratio of the area of overlap to the area of union, and its normalized average value is measured between 0 and 1. MAP is the mean of the average precisions (APs) over a complete dataset. Recall is the metric that specifies the ability of the object detector to successfully detect the object of interest, i.e., the ratio of true positives to the total number of cases. Table 3, Table 4 and Table 5 are the based-on precision parameters and clearly depict the performance of the proposed algorithm was better than its counterparts in terms of precision, MAP and recall. The algorithms are tested on a Corei7 machine in a MATLAB 2019 environment to maintain uniformity.

## 4. Conclusions

The paper presents a tracking routine that tracks the object of interest (such as an object or a subject) using the gradient descent approach. The algorithm detects an object using the MACH filter, which recognizes the object of interest using ASM and generates a correlation peak depicting the presence of an object. A bounding box is constructed around the object once MACH recognizes it. The coordinates of the bounding box are then fed to the gradient descent-based object tracking routine, which tracks the object of interest in successive frames using a step size and the terminal function. The presence of the terminal function enables a much faster convergence of particles in the gradient-based algorithm compared to the conventional state-of-the-art algorithms. The proposed algorithm has a significant scope to improve in the future, as the gradient descent algorithm can be further improved and its ensemble with particle swarm optimization can yield even better convergence results for object and human recognition [49,50,51]. Moreover, the deep learning based shall be more useful for the recognition task [52,53,54,55,56,57,58,59].

## Figures and Tables

**Figure 1 sensors-22-01098-f001:**
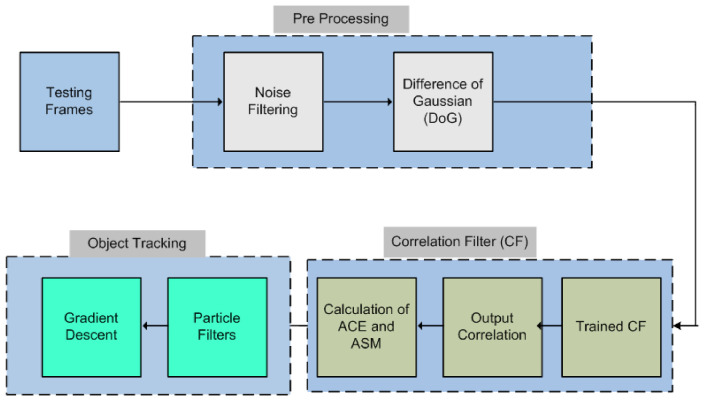
Proposed model of a system.

**Figure 2 sensors-22-01098-f002:**
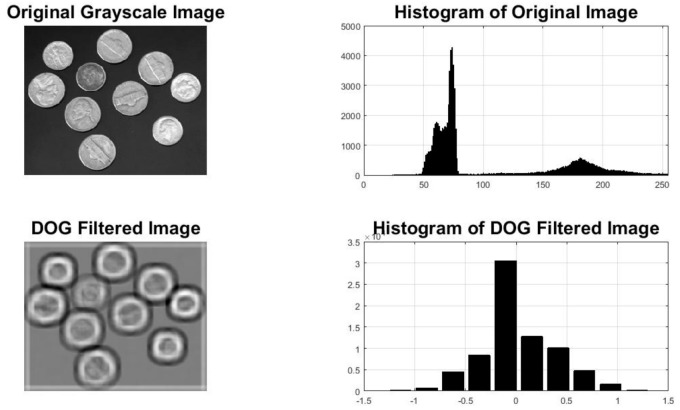
Visual explanation of the difference of Gaussian (DoG) method.

**Figure 3 sensors-22-01098-f003:**
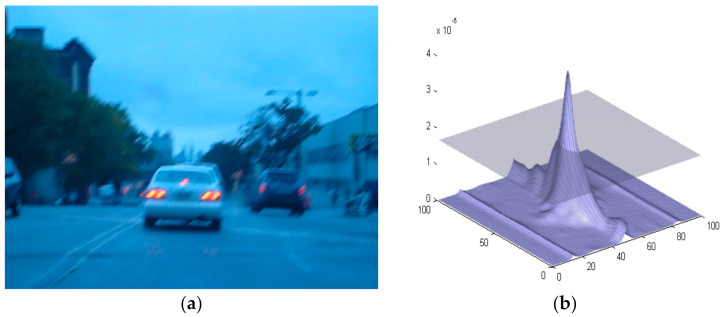
(**a**) A sample image of Blur Car (Data Set-1) and (**b**) MACH results.

**Figure 4 sensors-22-01098-f004:**
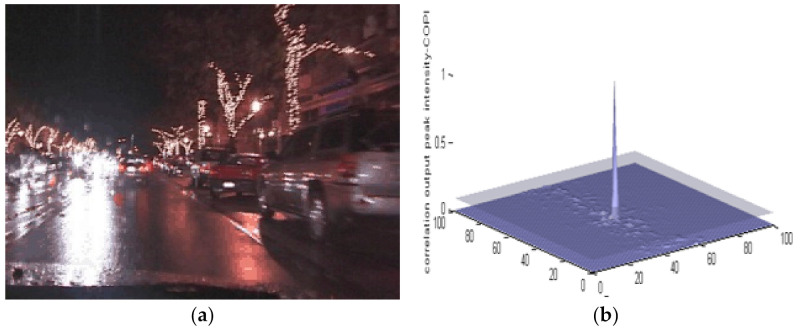
(**a**) Vehicle traveling at night (Data Set-3) (**b**) MACH results.

**Figure 5 sensors-22-01098-f005:**
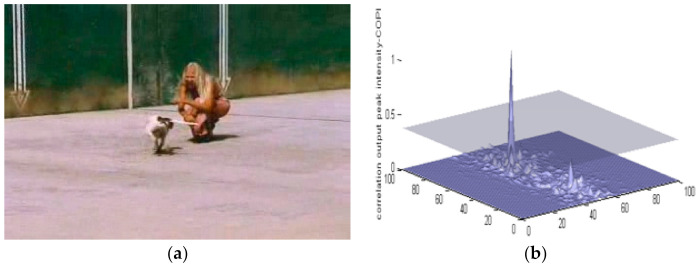
(**a**) A sample image of a Running Dog (Data Set-2) and (**b**) MACH results.

**Figure 6 sensors-22-01098-f006:**
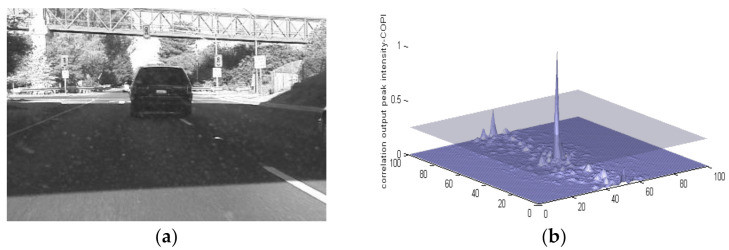
(**a**) An occluded grayscale vehicle (Data Set-4) and (**b**) MACH results.

**Figure 7 sensors-22-01098-f007:**
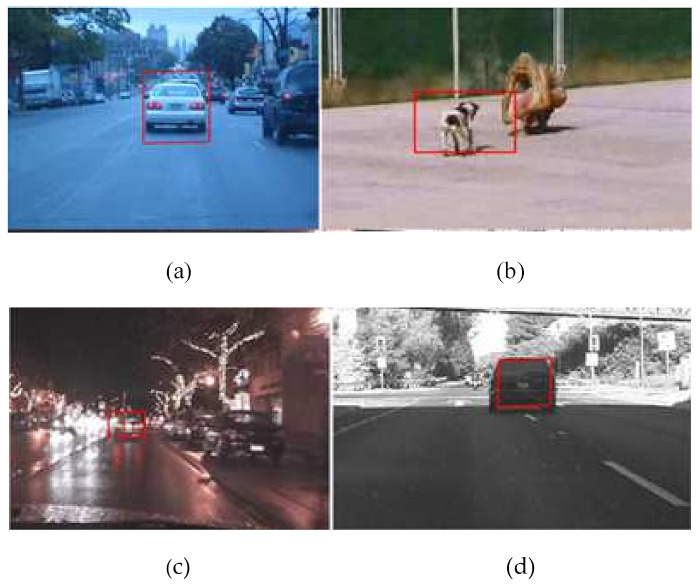
Detected objects for sample images: (**a**) Blurred Vehicle; (**b**) Running Dog; (**c**) Vehicle at Night; (**d**) Car moving in a lane.

**Figure 8 sensors-22-01098-f008:**
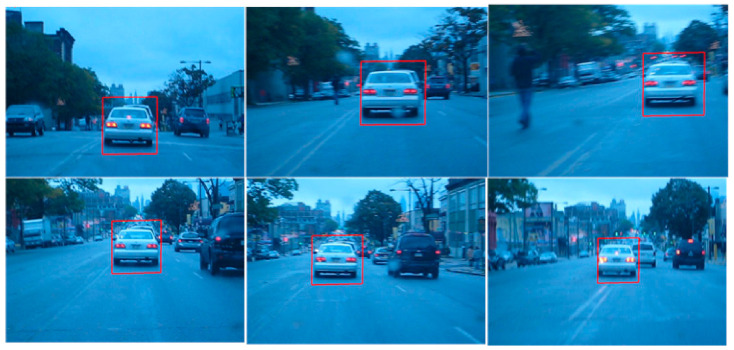
Tracking of Vehicle (Data Set: Blur Car).

**Figure 9 sensors-22-01098-f009:**
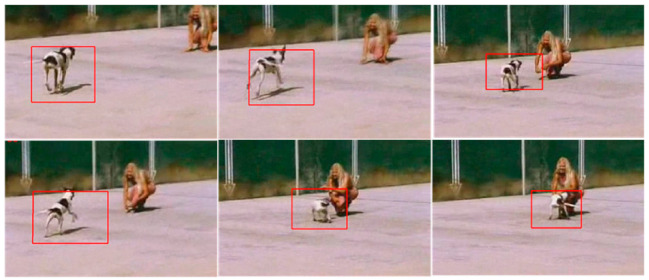
Tracking of a running dog (Data Set: Running Dog).

**Figure 10 sensors-22-01098-f010:**
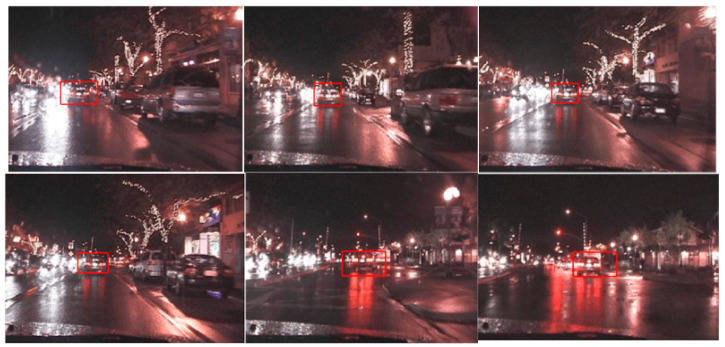
Tracking of Vehicle (Data Set: Vehicle at Night).

**Figure 11 sensors-22-01098-f011:**
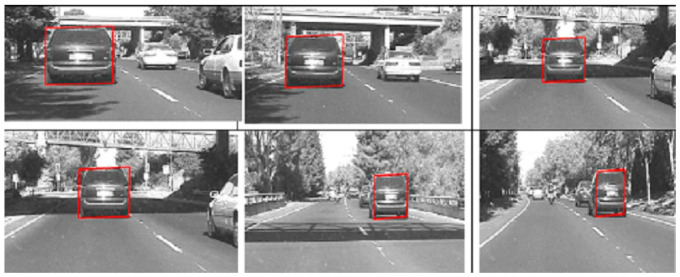
Tracking of a grayscale occluded vehicle (Data Set: Grayscale Vehicle).

**Figure 12 sensors-22-01098-f012:**
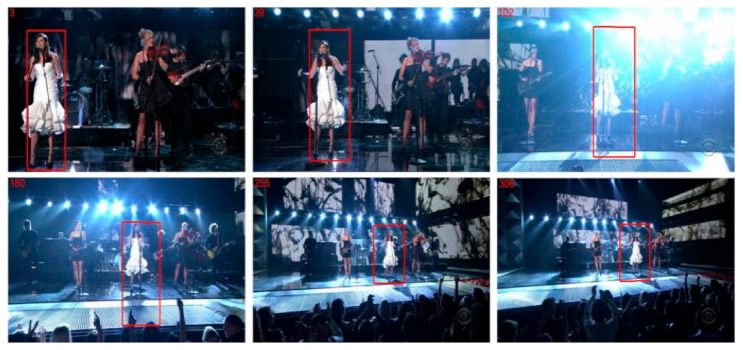
Tracking of a human person in a complex environment (Data Set: Singer).

**Table 1 sensors-22-01098-t001:** Comparison of state-of-the-art algorithms in terms of execution time (sec.).

Comparison of Execution Time (in Seconds) of Algorithms (Min. 300 Frames)							
Data Set	TTACNN	ADT	VTACDT	APGCF	AWSODM	MTUMT	Proposed Algorithm
Blur Car	2.14	2.51	2.10	2.44	2.91	2.29	2.01
Running Dog	2.92	4.12	2.77	4.11	2.99	2.84	2.89
Vehicle at Night	3.04	3.09	2.91	3.19	2.71	2.69	2.72
Grayscale vehicle	2.46	3.01	2.62	2.90	2.19	2.90	2.21
Singer	2.99	3.71	2.81	2.81	2.89	3.11	2.85

**Table 2 sensors-22-01098-t002:** Comparison of different techniques based on Average Tracking Errors.

Average Tracking Errors (Min. 300 Frames)							
Data Set	TTACNN	ADT	MTUMT	VTACDT	APGCF	AWSODM	Proposed Algorithm
Blur Car	0.46	0.42	0.21	0.17	0.055	0.21	0.041
Running Dog	0.059	0.057	0.48	0.056	0.051	0.061	0.048
Vehicle at Night	0.09	0.088	0.099	0.094	0.071	0.041	0.012
Grayscale vehicle	0.10	0.101	0.118	0.089	0.09	0.088	0.079
Singer	0.14	0.14	0.211	0.1328	0.129	0.144	0.127

**Table 3 sensors-22-01098-t003:** Performance evaluation based on Precision.

Comparison Based on Precision (Min. 300 Frames)							
Data Set	TTACNN	ADT	MTUMT	VTACDT	APGCF	AWSODM	Proposed Algorithm
Blur Car	0.88	0.88	0.93	0.94	0.94	0.92	0.96
Running Dog	0.90	0.92	0.89	0.93	0.94	0.87	0.94
Vehicle at Night	0.91	0.95	0.88	0.92	0.97	0.86	0.98
Gray scale vehicle	0.96	0.96	0.91	0.97	0.99	0.97	1.00
Singer	0.94	0.92	0.95	0.98	0.97	0.98	1.00

**Table 4 sensors-22-01098-t004:** Performance evaluation based on MAP.

Comparison Based on MAP (Min. 300 Frames)							
Data Set	TTACNN	ADT	MTUMT	VTACDT	APGCF	AWSODM	Proposed Algorithm
Blur Car	69.6	64.8	69.9	73.9	70.1	73.8	74.6
Running Dog	74.0	66.9	71.0	73.1	71.9	71.7	72.9
Vehicle at Night	77.2	68.2	71.1	76.9	74.6	75.1	77.8
Gray scale vehicle	76.1	69.2	72.5	74.9	77.1	77.9	78.2
Singer	74.9	66.0	72.8	75.5	70.9	72,8	75.6

**Table 5 sensors-22-01098-t005:** Performance evaluation based on Recall.

Comparison Based on Recall (Min. 300 Frames)							
Data Set	TTACNN	ADT	MTUMT	VTACDT	APGCF	AWSODM	Proposed Algorithm
Blur Car	0.55	0.52	0.59	0.53	0.59	0.55	0.52
Running Dog	0.52	0.54	0.54	0.45	0.54	0.49	0.45
Vehicle at Night	0.46	0.44	0.39	0.46	0.49	0.44	0.41
Gray scale vehicle	0.49	0.46	0.46	0.42	0.51	0.44	0.40
Singer	0.41	0.38	0.44	0.44	0.39	0.39	0.35

## Data Availability

The Urban Lisa dataset is available from http://homepages.inf.ed.ac.uk/rbf/CVonline/Imagedbase.htm (accessed on 11 October 2021). The Visual Tracker Benchmark is available from http://cvlab.hanyang.ac.kr/tracker_benchmark/datasets.html (accessed on 11 October 2021).

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
