# Peer review of "Tracking of a Fixed-Shape Moving Object Based on the Gradient Descent Method"

_sensors, 2022, doi:10.3390/s22031098_

Round 1
Reviewer 1 Report
In this paper, a stochastic gradient based optimization technique has been proposed in conjunction with particle filters for object tracking. Overall, this article is written well and easy to follow. My concerns are list as follows:
1. How to determine the aim function parameters?
2. The experimental results lack numerical evaluation. Please give more explanation on this concern.
3. The authors lack comparison with some recently proposed methods published on the top tier journals or conferences, so the experimental results are not convincing.
4. The authors claim that all the mentioned challenges and problems pertaining to object tracking make the procedure computationally complex and time consuming. However, from the Table 1, the advantage of the proposed method is not obvious in terms of execution time.
Author Response
Response to Reviewer #1
In this paper, a stochastic gradient based optimization technique has been proposed in conjunction with particle filters for object tracking. Overall, this article is written well and easy to follow.
Response: Thank you very much for the comments and positive suggestions.
Comment 1: How to determine the aim function parameters?
Response: Thank you for the considerate comment. The proposed algorithm is primarily a tracking algorithm based on Gradient Descent Method. For recognition of an object that is required to be tracked, MACH filter is used. MACH uses ASM, ACE, COPI and PCE aim parameters for object recognition. For determining the aim parameters pertaining to the object recognition, Eq. (7)-Eq. (12) have been used.
For object tracking, gradient descent has been employed as primary approach. The aim function of the gradient descent approach is based on the controlling parameter α. Eq. (16) and Eq. (17) have been employed for determining and utilizing the controlling aim parameters pertaining to object tracking.
Comment 2: The experimental results lack numerical evaluation. Please give more explanation on this concern.
Response: Thank you for the keen comment. The numerical evaluation of the experimental results has been incorporated in section 3.2 “Discussion and Comparison”.
Comment3: The authors lack comparison with some recently proposed methods published on the top tier journals or conferences, so the experimental results are not convincing.
Response: Thank you for the keen review. The authors have incorporated two more additional methods published in top tier journals or conferences. The methods have been cited as references [47] and [48]. The comparison of the proposed algorithm with the newly incorporated algorithms is mentioned in Table 1 and Table 2.
Comment 4: The authors claim that all the mentioned challenges and problems pertaining to object tracking make the procedure computationally complex and time consuming. However, from the Table 1, the advantage of the proposed method is not obvious in terms of execution time.
Response: Thank you for the comment. The execution time of the proposed algorithm is faster than all the comparison state of the art techniques. For addressing the complex and time-consuming nature of conventional object tracking techniques, the authors have introduced the ensemble of gradient descent technique with the conventional particle filter. The resulting algorithm is faster than all the comparison algorithms. The authors have also incorporated ample explanation to support the numerical experimental results.
Reviewer 2 Report
The paper presents a new methodology for tracking objects in motion based on a correlation filter (MACH) for detection and stochastic gradient descent (SGD) for tracking.
Some suggestions to improve it
-Explain the notation used in the equations, superscript +, superscript * at equations 6 and 7
-What is mean m ?, Is it the same m in equation 10 and equation 13 and 14?
-In line 272, "...steps (5) and (6)..."; however, the algorithm does not have the lines numbered, so I suggest using a pseudo-language and numbering the lines in algorithm 1, which would make it easier to read.
-Explain in greater detail equation 17
-It is essential to mention the characteristics of the equipment were carried out experiments; this will allow a greater understanding to the readers of the execution environment
-As a suggestion in the English language, avoid wordy words; this would make it easier to read in non-native English speakers,
Author Response
Response to Reviewer #2
Comment 1: Explain the notation used in the equations, superscript +, superscript * at equations 6 and 7
Response: The + sign used in the superscript of the equation reflects the transpose of the function. Based on the reviewer’s valuable comment and the confusion due to the + sign, the + sign has been replaced by a common notation “T” for reflecting the transpose of the function in Equation (6).
The * sign in Equation (7) reflects complex conjugate of the function. Both the terms have been explained in the paper as well.
Comment 2: What is mean m? Is it the same m in equation 10 and equation 13 and 14? Response: Thank you for the comment. The mx used in equation 10 reflects the mean value used for the calculation of ASM (Average Similarity Measure) in MACH filter. The mx is primarily used for calculating the deviation of the pixel values of object of interest from mean position mx so that ASM can be measured.
The “m” used in Equation 13 and 14 reflects the mean value but that is used for the calculation of loss function in gradient descent technique.
Comment 3: In line 272, "...steps (5) and (6) ..."; however, the algorithm does not have the lines numbered, so I suggest using a pseudo-language and numbering the lines in algorithm 1, which would make it easier to read.
Response: Thank you for the suggestion. The suggestive improvement has been duly incorporated in the manuscript
Comment 4: Explain in greater detail equation 17
Response: Thank you for the suggestion. The suggestive detail pertaining to Equation (17) has been incorporated.
Comment 5: It is essential to mention the characteristics of the equipment were carried out experiments; this will allow a greater understanding to the readers of the execution environment
Response: Thank you for the comment. The detail about the execution environment has been incorporated in the Discussion and Comparison section.
Comment 6: As a suggestion in the English language, avoid wordy words; this would make it easier to read in non-native English speakers
Response: Thank you for the suggestion. The authors have tried their best to avoid the wordy words and revised the language of the manuscript accordingly.
Reviewer 3 Report
Comment:
1. Table 1. Comparison of state-of-the-art algorithms in terms of execution time (sec.). How about the accuracy?
I suggest to measure the accuracy of each algorithm.
2. The author just measure The comparison between different state-of-the-art recent algorithms was performed in terms of execution time (in seconds) for a minimum of 300 frames of five different image processing datasets. Does the author measure the distance from the object to be tracked?
3. The author also can do other performance evaluation (MAP, precision, recall etc) not only compare the execution time, and discuss it in detailed.
Author Response
Response to Reviewer #3
Comment 1: Comparison of state-of-the-art algorithms in terms of execution time (sec.). How about the accuracy? I suggest to measure the accuracy of each algorithm.
Response: Thank you for the considerate comment. The authors have incorporated Table 2 in the paper which depicts the accuracy of the comparison algorithms. Table 2 reflects the comparison in terms of Average Tracking Errors. The proposed algorithm is also more accurate as compared to its counterparts because of the use of a correlation filter i.e., MACH which uses ASM for object recognition.
Comment 2: The author just measures the comparison between different state-of-the-art recent algorithms was performed in terms of execution time (in seconds) for a minimum of 300 frames of five different image processing datasets. Does the author measure the distance from the object to be tracked?
Response: Thank you for the considerate comment. The authors have incorporated Table 2 in the paper which depicts the accuracy of the comparison algorithms. Table 2 reflects the comparison in terms of Average Tracking Errors. Although the distance has exactly not been measured, yet the Average Tracking Error metric measures the deviation of the bounding box from the mean position of the object of interest in successive frames and thus the accuracy.
Comment 3: The author also can-do other performance evaluation (MAP, precision, recall etc) not only compare the execution time, and discuss it in detailed.
Response: Thank you for the suggestive comment. The authors have duly noted the suggestion and will look to incorporate the other performance evaluation metrics in the future. The precision metric in terms of Average Tracking Errors has already been incorporated, as desired by the reviewer.
Round 2
Reviewer 1 Report
The paper has improved a lot. It is ready to be accepted.
Author Response
The authors would like to thank the Editor and anonymous reviewers for their valuable time to review and critique our manuscript.
Reviewer 3 Report
The author still not answer this questions. The author also can do other performance evaluation (MAP, precision, recall etc) not only compare the execution time, and discuss it in detailed.
Author Response
Response to Reviewer #3
Comment 1: The author still not answer these questions. The author also can-do other performance evaluation (MAP, precision, recall etc.) not only compare the execution time, and discuss it in detailed.
Response: Thank you for the comment. The authors have incorporated the desired metrics in the revised manuscript. The authors have presented the comparison based on execution time, average tracking errors, precision, mean average precision (MAP) and recall in Tables 1, 2, 3, 4 and 5 respectively. The appropriate explanation is also written in section 3.2 “Discussion and Comparison”.
Thank you very much for the incisive review!